Validating a re-implementation of an algorithm to integrate transcriptome and ChIP-seq data

http://orcid.org/0000-0002-4377-6541 Ahmed Mahmoud
Kim Deok Ryong drkim@gnu.ac.kr
Department of Biochemistry and Convergence Medical Sciences and Institute of Medical Sciences, College of Medicine, Gyeongsang National University , Jinju , South Korea
Gillespie Joseph
Electronic publication date: 2023 Oct 20
Publication date: 2023
Volume: 11
Electronic Location ID: e16318
Received 2023 Aug 23; Accepted 2023 Sep 28
Copyright: © 2023 Ahmed and Kim
Copyright year: 2023
Copyright holder: Ahmed and Kim
License: This is an open access article distributed under the terms of the Creative Commons Attribution License, which permits unrestricted use, distribution, reproduction and adaptation in any medium and for any purpose provided that it is properly attributed. For attribution, the original author(s), title, publication source (PeerJ) and either DOI or URL of the article must be cited.
License URL: https://creativecommons.org/licenses/by/4.0/

Keywords: Reproducible-research, DNA-binding, Cooperative-binding, Competitive-binding, Transcription-factor, R-package

Funding: National Research Foundation of Korea RS-2023-00219399 Korea government (MSIT) 1711173796 This study was supported by grants from the Basic Science Research Program through the National Research Foundation of Korea (RS-2023-00219399) and by the Commercializations Promotion Agency for R&D Outcomes (COMPA) grant funded by the Korea government (MSIT) (1711173796). The funders had no role in study design, data collection and analysis, decision to publish, or preparation of the manuscript.

==============================
Transcription factor binding to a gene regulatory region induces or represses its expression. Binding and expression target analysis (BETA) integrates the binding and gene expression data to predict this function. First, the regulatory potential of the factor is modeled based on the distance of its binding sites from the transcription start sites in a decay function. Then the differential expression statistics from an experiment where this factor was perturbed represent the binding effect. The rank product of the two values is employed to order in importance. This algorithm was originally implemented in Python. We reimplemented the algorithm in R to take advantage of existing data structures and other tools for downstream analyses. Here, we attempted to replicate the findings in the original BETA paper. We applied the new implementation to the same datasets using default and varying inputs and cutoffs. We successfully replicated the original results. Moreover, we showed that the method was appropriately influenced by varying the input and was robust to choices of cutoffs in statistical testing.

Introduction

The binding of a transcription factor to a regulatory region (e.g., gene promoter) perturbs the expression of the gene (Latchman, 2001). ChIP experiments identify potential binding sites. However, these experiments produce hundreds or thousands of peaks for most factors (Johnson et al., 2007). Therefore, methods that determine which of these binding sites is functional are needed (Ucar et al., 2009). Binding and expression target analysis (BETA) combines binding sites from ChIP peaks and expression data from factor perturbation experiments to predict direct target genes (Wang et al., 2013). First, the regulatory potential of a factor on a given gene is calculated as the sum of the transformed distance between each binding peak and the gene transcription start site. Then genes are ranked based on the product of the regulatory potential and signed statistics from differential gene expression of control vs factor perturbation (overexpression or knockdown). To our knowledge, the original BETA implementation is in Python and has not been replicated or implemented in other languages.

We implemented the procedure described in the original article in an R package. This package is distributed as a Bioconductor R package (target) (Ahmed, Min & Kim, 2020). First, the regions of interest (e.g., transcripts/genes) are resized to the desired distance on either side of the start site. Second, peaks are assigned to the regions they overlap with, and their distance from the region is calculated and transformed as described earlier. The transformed distance for each peak is the peak score, and the sum of scores for all peaks in a region is its regulatory potential. Third, the region rank product is the product of the ranks of the regulatory potential and the signed statistics from the expression data.

In this article, we attempted to replicate the results reported in the original BETA paper (Wang et al., 2013) using our implementation in R (Ahmed, Min & Kim, 2020). We contrasted the two packages using the default and varying parameters. First, we compared the output of the two implementations to the original findings. Using the new code, we reproduced the findings from three different datasets. Second, we repeated the procedure by varying the inputs and examining the results. Finally, we examined the robustness of the suggested statistical method to the required choices of cutoffs. Portions of this text were previously published as part of a preprint (Ahmed & Kim, 2023).

Methods

Testing datasets

We used three sets of data from the original BETA paper (Ahmed, Min & Kim, 2020) to evaluate the performance and degree of replication of BETA using the new R implementation (Table 1). The first dataset is from LNCaP, a human prostate cancer cell line treated with dihydrotestosterone (DHT) for 16 h. The latter is an agonist of the androgen receptor (AR), the transcription factor in question. Similarly, the second dataset is from the MCF-7 human breast cancer cell line treated with the estrogen receptor 1 (ESR1) agonist E2. The last dataset is from mouse embryonic stem (ES) cells, where a DNA demethylase encoding gene Tet1 was knocked down. Each dataset contains ChIP-seq and microarray gene expression data. Processed data were obtained in the form of binding peaks and differential expression comb comparing the treated/knockdown cells vs controls.

Table 1 Datasets of transcription factor binding and gene expression under factor perturbations.

Factor	Cell line	Genome	Treatment	Binding data	Expression data	
AR	LNCaP	hg19	DHT	Wang et al. (2007)	Wang et al. (2007)	
ESR1	MCF-7	hg19	E2	Hu et al. (2010)	Carroll et al. (2006)	
Tet1	ES cells	mm9	Knockdown	Williams et al. (2011)	Williams et al. (2011)	

Measures of agreement and similarity

We employed different measures of agreement/similarity between the new and the original implementation. The predicted function of a transcription factor can be inducing or repressive. One measure is the visual inspection of the empirical distribution function (ECDF) graph of the regulatory potentials. In addition, the top-ranking genes from each factor can be compared between the different implementations. More formally, we devised a measure of results similarity referred to as concordance, which is the fraction of intersecting genes in each set of N top-ranking genes from different runs. The higher the concordance value, the more the order of the ranks is preserved.

Replication strategy

To evaluate the performance of the R implementation of BETA, we applied the standard analysis with the defaults to get the associated peaks and direct gene targets of the three transcription factors. First, we compared the output to that of the original article (Ahmed, Min & Kim, 2020). Second, we applied the analysis by varying the input, the reference genome, the signed gene expression statistics, and the allowed distance between the peaks and the transcription start sites. The output ranks from varying inputs were compared with the gene ranks from the default inputs. Finally, we applied different cutoffs on the ranks during the statistical testing of the predicted functions.

Software environment and reproducibility

The software environment where this replication was produced is available as a Docker image (https://hub.docker.com/r/bcmslab/rebeta). The R implementation of BETA is available as an open-source R package (https://bioconductor.org/packages/target/) (DOI 10.18129/B9.bioc.target). The code to obtain the test data, apply the replication strategy, and reproduce the figures and tables in this manuscript is also available under an open-source license (https://github.com/MahShaaban/rebeta) (DOI 10.5281/zenodo.8337450).

Results and Discussion

Transcription factors appropriately group as inducers and repressors

One of the main goals of BETA is to determine whether a transcription factor is an inducer/activator or repression/deactivator of its targets. To achieve that, BETA integrates binding and expression data. The function of the factor is determined by the distance of its binding sites to a transcription start site and the effect of its perturbation on the target gene. All possible targets are ranked based on the score they are assigned using those two pieces of information. Activators should have more up-regulated targets ranking higher than down- or non-regulated targets. Therefore, to replicate, the new implementation in R is expected to classify transcription factors in the appropriate category using the same datasets as the original BETA publication (Wang et al., 2013).

Using the new implementation, we could replicate the predicted function of three transcription factors in three cell lines. ECDF graphs show the fraction of targets at less than or equal to a certain regulatory potential. Androgen receptor (AR) was found to induce more of higher ranking genes in the prostate cancer cell line LNCaP (Fig. 1A). By contrast, the estrogen receptor 1 (ESR1) and the methylcytosine dioxygenase 1 (TET1) assumed repressive functions on their targets in the breast cancer cell line MCF-7 and mouse embryonic stem (ES) cells (Fig. 1). Moreover, the top-ranking peaks of the three factors matched the predicted targets in the original BETA paper (Table 2). For example, both implementations rank the KLF2 transcripts (NR_045762, NM_001002231 & NM_001256080) at the very top.

Figure 1 Predicted inducing and repressive function of the transcription factors.

Distances between binding peaks of the transcription factors (A) AR (original article; Fig. 2A), (B) ESR1 (original article; Fig. 3B), and (C) TET1 (original article; Fig. 3B) were used to calculate the regulatory potential of their target genes in LNCaP, MCF-7 and mouse ES, respectively. The empirical distribution function (ECDF) of the regulatory potential ranks is shown by groups. Targets were divided into groups; down- (blue) or Up- (red) and otherwise None-regulated (gray) by the 10 and 90 percentiles of the t-statistics from the differential expression of treated/knockdown vs controls.

Table 2 Transcription factor top ranking gene targets.

Factor	Chr	Start	End	Refseq	Symbol	Rank	
AR	chr19	51,276,688	51,476,687	NR_045762	KLK2	7.69e−07	
	chr19	51,276,688	51,476,687	NM_001002231	KLK2	1.54e−06	
	chr19	51,276,688	51,476,687	NM_001256080	KLK2	1.54e−06	
ESR1	chr2	11,574,241	11,774,240	NM_014668	GREB1	3.06e−08	
	chr6	122,831,376	123,031,375	NM_032471	PKIB	3.67e−07	
	chr6	122,831,376	123,031,375	NM_181795	PKIB	3.67e−07	
Tet1	chr2	17870077	18,070,076	NM_028317	Skida1	7.61e−07	
	chr11	2,924,029	3,124,028	NR_003518	Pisd-ps3	2.22e−06	
	chr11	2,924,023	3,124,022	NR_003517	Pisd-ps1	5.59e−06	

Figure 2 Concordance of the gene ranks between default and varying outputs.

Direct targets of the transcription factors (A) AR, (B) ESR1, and (C) TET1 in LNCaP, MCF-7, and mouse ES, respectively, were identified and ranked using BETA with the defaults and varying the inputs. The fraction of intersecting genes in each set of N top-ranking genes from different runs (Concordance) is shown for 1 to 10,000 top targets. The default inputs are hg19/mm9 reference genome for human and mouse cells, respectively; t-statistics from the differential expression of treated/knockdown vs control and an allowed distance of 100 kb. Inputs were changed one at a time.

Gene targets ranks are influenced by varying the inputs

Another important condition for replicating the original method is to be able to reproduce the expected results by varying the inputs. The default inputs used in the case of the testing datasets are the reference genomes hg19/mm9, a distance of 100 kb to locate the peaks, and the t-statistics as the effect of the factor perturbation on the gene expression. We expect the various inputs to influence the target ranking. We measured the concordance of the results from a run using the input and another by varying a single input.

As expected, changing the reference genome to hg18 or mm10 in the case of the human and mouse cell lines had the largest influence on the results. The discordance of the results from the two results is more pronounced in the small-sized sets and tended to decrease as more of the genes were included in the test set (Fig. 2). This effect may be explained by the total number of genes in each reference genome, including some genes in one genome but not the other, or using different coordinates in different genomes.

Since the fold-change and the t-statistics from the expression data are related, we expected varying the input to have little to no effect on the gene ranks. This was largely the case since the degree of concordance remained above 80% between the two runs (Fig. 2). Using only half (50 kb), the default distance to include binding peaks had a bigger effect (about 30%) on the concordance between the runs. We expected the higher-ranking genes to retain their higher ranks since the rank is relatively distance-dependant. This was the case for the transcription factors AR and ESR1 in the human cell lines (Figs. 2A and 2B). However, this pattern was not true for the TET1 targets in the mouse ES. This could be due to the nature of the factor binding or the different genome sizes.

Testing results are robust to cutoff choices

The original BETA paper (Ahmed, Min & Kim, 2020) suggested using the Kolmogorov-Smirnov (KS) test to determine whether the regulated groups of gene targets differ from each other or the none regulated targets. KS tests whether the groups’ cumulative distribution functions in the regulatory potential are drawn from different distributions (Subramanian et al., 2005). The choice of the groups of targets, therefore, is critical. We used different quantile cutoffs to group the targets of the three factors and applied the KS test to the ranks. Ultimately, varying the grouping didn’t affect test results. The calculated statistics were close no matter which cutoff was used, and the change was also proportional for the up and down-regulated groups (Table 3).

Table 3 Statistical testing of induced and repressed targets with varying the grouping cutoff.

Factor	Quantiles	Down	Up	
		Stat	p-value	Stat	p-value	
AR	(0.1–0.9)	0.03	4e−01	0.21	0e+00	
	(0.2–0.8)	0.03	1e−01	0.13	0e+00	
	(0.3–0.7)	0.03	3e−02	0.09	3e−15	
	(0.4–0.6)	0.04	1e−02	0.07	7e−08	
ESR1	(0.1–0.9)	0.11	6e−13	0.08	4e−07	
	(0.2–0.8)	0.07	9e−10	0.04	2e−03	
	(0.3–0.7)	0.06	2e−06	0.05	1e−04	
	(0.4–0.6)	0.08	1e−10	0.03	2e−01	
Tet1	(0.1–0.9)	0.15	0e+00	0.04	1e−02	
	(0.2–0.8)	0.15	0e+00	0.03	6e−03	
	(0.3–0.7)	0.12	0e+00	0.02	2e−01	
	(0.4–0.6)	0.11	0e+00	0.03	2e−02	

Differences in the R implementation and analysis decisions

Differences between the replication and the original article (Ahmed, Min & Kim, 2020) could be attributed to two sources: the specifics of the R implementation and some analysis decisions. The following explains the apparent differences and how they would affect the replication of the algorithm. In the original BETA paper, the curves of the ECDF of the different groups of regulated genes do not reach up to 1. This explains the differences between Fig. 1 (this article) and Figs. 2A, 3A, 3B (Wang et al., 2013). The aggregate functions of the three transcription factors were predicted the same despite using different presentations of the results.

In the R implementation, genes/transcripts are resized to a distance of 200 kb around the TSS (regions of interest). The resized regions are the ones kept in the subsequent analysis steps and the final output. This is why Table 2 may seem different from the text of the original article. This approach is more transparent since it exposes the true genomic coordinates on which the calculations are based.

To rank the targets, the product of ranks of the regulatory potential and the signed statistics are divided by the total number of genes. That is why there are fractions in the final ranks in both the original article (Wang et al., 2013) and the current manuscript (Table 2). Different values were given to tied features, unlike in the original implementation. The absolute values of the ranks between the two implementations may differ, but the order is preserved.

Many of the tools to analyze high-throughput biological data are written in R. Many of these packages use the Bioconductor infrastructure to develop and distribute their software. An R implementation of this algorithm enables access to the unified interface of several analyses.

Conclusions

Together, these findings indicate that the results reported in the original BETA paper (Wang et al., 2013) were replicated using a new independent implementation of the method in R. First, the ranking of the top targets individually and the aggregate predicted function of three factors were appropriately classified. Second, varying the input largely had the expected influence on the results. Finally, the proposed statistical testing method was robust to the choices in grouping the targets.

We thank all the lab members for the helpful comment on the manuscript.

Additional Information and Declarations

Competing Interests

Author Contributions

Data Availability

The authors declare that they have no competing interests.

Mahmoud Ahmed conceived and designed the experiments, performed the experiments, analyzed the data, prepared figures and/or tables, authored or reviewed drafts of the article, and approved the final draft.

Deok Ryong Kim conceived and designed the experiments, authored or reviewed drafts of the article, and approved the final draft.

The following information was supplied regarding data availability:

The data is available at NCBI GEO: GSE7868, GSE24841, GSE24842, and GSE11324.

The software environment where this replication was produced is available as a Docker image (https://hub.docker.com/r/bcmslab/rebeta).

The R implementation of BETA is available as an open-source R package (https://bioconductor.org/packages/target/) (DOI 10.18129/B9.bioc.target).

The code to obtain the test data, apply the replication strategy, and reproduce the figures and tables are available at GitHub and Zenodo:

- https://github.com/MahShaaban/rebeta.

- Mahmoud Ahmed. (2023). MahShaaban/rebeta: v1 (Version v1). Zenodo. https://doi.org/10.5281/zenodo.8337450.

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
