# Peer review of "Validating a re-implementation of an algorithm to integrate transcriptome and ChIP-seq data"

_PeerJ, doi:10.7717/peerj.16318_

## Round 0.1 · original submission · Major Revisions

Dear Drs. Ahmed and Kim:

Thanks for submitting your manuscript to PeerJ. I have now received two independent reviews of your work, and as you will see, the reviewers raised some concerns about the research. Despite this, this the reviewers are optimistic about your work and the potential impact it will have on research utilizing tools for transcription and gene regulation analyses. Thus, I encourage you to revise your manuscript, accordingly, taking into account all of the concerns raised by both reviewers.

While the concerns of the reviewers are relatively minor, this is a major revision to ensure that the original reviewers have a chance to evaluate your responses to their concerns.

Importantly, please ensure your Materials and Methods are clearly stated. The methods should be clear, concise and repeatable. Please ensure this, and make sure all relevant information and references are provided. Overall, the clarity and presentation of the paper needs improvement.

I agree with many of the concerns of the reviewers, and thus feel that their suggestions should be adequately addressed before moving forward.

I look forward to seeing your revision, and thanks again for submitting your work to PeerJ.

Good luck with your revision,

-joe

Reviewer 1 ·

Basic reporting

no comment

Experimental design

no comment

Validity of the findings

no comment

Additional comments

In the manuscript entitled Re-implementation of an algorithm to integrate transcriptome and ChIP-seq data’, Ahmed et al reimplemented the algorithm of binding and expression target analysis (BETA), which was originally implemented in Python, in R. This algorithm combines binding sites from ChIP peaks and expression data from factor perturbation experiments to predict direct target genes. Using the new implementation in R, the authors showed that they were able to reproduce the results reported in the original BETA paper.

I only have a few minor points for revision.

1. One major task the authors have done in the manuscript was to compare the current results to those reported in the original BETA paper. In some figures and tables, the authors should consider including results from the original paper (i.e., adding the top-ranking gene targets from the original paper in Table 2) so that it would be easier for the readers to compare and understand.
2. Could the authors add some discussions regarding the motivations/advantages of the re-implementation in R, presumably this can be things like it’s more compatible with some specific upstream/downstream pipelines/analyses?
3. Although it’s not required, could the authors include some comparisons regarding things like running/execution time and required computational resources between the previous and new implementations?

·

Basic reporting

The figures & tables are clear, and data & code are provided for reproducibility. However, I have some concerns over how the goal and impact of the paper is communicated in the current manuscript. Please see my suggestions below:

Suggested Improvements:
The title of this manuscript gives the impression that this is a new R package that re-implements an algorithm that was originally written in Python. However, the authors have already published this R package in a previous paper that they cite [1]. This current manuscript is instead an application of the existing R package to datasets used to test the original Python implementation. I suggest the authors make this clear by re-phrasing the title along the lines of "Validating target: an R package to integrate transcriptome and ChIP-seq data".

Along the same lines, I suggest removing the phrasing "a new implementation", such as in l24 of the manuscript, since this was already published in [1].

There are sections of the Introduction that are directly reproduced from the authors’ earlier published work [1], which is not ideal:

For example, here is the opening paragraph of this manuscript: "The binding of a transcription factor to a gene regulatory region (e.g., gene promoter) can have the effect of inducing or repressing its expression. Potential binding sites can be identified using ChIP experiments as the enriched regions within aligned reads (peaks). High throughput ChIP experiments produce hundreds or thousands of peaks for most factors. Therefore, methods to determine which of these sites are true targets and whether they are functional or not are needed. On the other hand, perturbing a transcription factor coding gene by overexpression or knockdown provides useful information on the function of the factor, such as by measuring the changes in gene expression. Methods exist to integrate the binding and the gene expression data from the factor perturbation to identify the real target regions (e.g., genes) and predict the transcription factor role."

And for comparison, here is the opening paragraph of the authors' previously published work: "The binding of a transcription factor to the regulatory region (e.g. gene promoter or enhancer) of a particular gene induces or represses its gene expression. High-throughput chromatin immunoprecipitation (ChIP) experiments identify hundreds or thousands of binding sites for most factors. Therefore, methods are needed to determine which of these sites are true targets and whether they are functional. Perturbing the transcription factor coding gene by overexpression or knockdown and measuring the effects on cellular gene expression provides useful information on the function of the factor. Methods exist to integrate binding and gene expression data of the factor perturbation to predict the direct target regions (e.g. genes)."

I would encourage the authors to remove or condense this section and shift the focus of the introduction to line 23 - line 29 which more clearly states the main goal of this manuscript, i.e. to replicate results from the original BETA paper using an existing R package. Relatedly, I would suggest moving the first paragraph of the Methods section: "Implementing BETA in R" to the Introduction section, since it refers to previous work.

Ref:
[1] M. Ahmed, D. S. Min, and D. R. Kim. “Integrating binding and expression data to predict transcription factors combined function”. BMC Genomics 21.1 (Dec. 2020), p. 610. ISSN: 1471-2164.

Experimental design

The experiment design is robust. The authors propose a measure of similarity called concordance, which is a good approach to compare different implementations, and the replication strategy is feasible and accurate. I appreciate the authors making available the software environment for testing and reproducibility.

Validity of the findings

The findings are in-keeping with previous work and are largely robust to parameter variation. I appreciate the authors analyzing the source of differences between their findings and that from the original paper, which is useful.

Additional comments

Generalizability: As mentioned above, since this manuscript tests the authors' previously developed tool on an existing dataset, it is challenging to make a strong case for the novelty of the paper either in method development or in new dataset availability. Therefore, I encourage the authors to instead split the Results and Discussions sections each into their own section and use the Discussion section to highlight some general learnings from their work that could be useful to future method developers.

---

## Round 0.2 · accepted · Accept

Dear Drs. Ahmed and Kim:

Thanks for revising your manuscript based on the concerns raised by the reviewers. I now believe that your manuscript is suitable for publication. Congratulations! I look forward to seeing this work in print, and I anticipate it being an important resource for groups studying tool development and utlization for transcription and gene regulation analyses. Thanks again for choosing PeerJ to publish such important work.

Best,

-joe

·

Basic reporting

I thank the authors for addressing my comments on clarifying the goal of the paper -- it is now clear that the paper seeks to validate the authors' R package on datasets used in the original BETA paper. Further, the authors have made suitable changes to the introduction & methods section that address my concerns regarding overlap with an earlier publication, and emphasize the novelty of the methods used in this paper.

Experimental design

No comment

Validity of the findings

No comment

Additional comments

No comment